# Development of a DSP Microcontroller-Based Fuzzy Logic Controller for Heliostat Orientation Control

**Eugenio Salgado-Plasencia, Roberto V. Carrillo-Serrano** **and Manuel Toledano-Ayala ***

División de Investigación y Posgrado, Facultad de Ingeniería, Universidad Autónoma de Querétaro,
Cerro de las Campanas s/n, Santiago de Querétaro, Querétaro 76010, Mexico; eugenio.salgado@uaq.mx (E.S.-P.);
roberto.carrillo@uaq.mx (R.V.C.-S.)
* Correspondence: toledano@uaq.mx; Tel.: +52-442-144-5820

**Abstract:** This paper describes the design and implementation of a heliostat orientation control system based on a low-cost microcontroller. The proposed system uses a fuzzy logic controller (FLC) with the Center of Sums defuzzification method embedded on a dsPIC33EP256MU806 Digital Signal Processor (DSP), in order to modify the orientation of a heliostat by controlling the angular position of two DC motors connected to the axes of the heliostat. The FLC is compared to a traditional Proportional-Integral-Derivative (PID) controller to evaluate the performance of the system. Both the FLC and PID controller were designed for the position control of the heliostat DC motors at no load, and then they were implemented in the orientation control of the heliostat using the same controller parameters. The experimental results show that the FLC has a better performance and flexibility than a traditional PID controller in the orientation control of a heliostat.

**Keywords:** heliostat; sun tracking; solar energy; embedded system; fuzzy logic control; center of sums defuzzification method

## 1. Introduction

The output power produced by a solar plant is proportional to the amount of solar energy absorbed by the system. Therefore, a sun tracking system (STS) with a high degree of accuracy is necessary to avoid losses in the output power of solar plants. STSs are usually classified into two categories [1]: passive sun tracking systems, which use the expansion of a gas caused by the solar radiation to move the mechanical structure of the tracker, and active sun tracking systems, which use motors, gears and electric controllers to drive concentration and absorption devices in a solar plant. There are two types of active sun tracking systems based on their controlling methods [2]: sensor driver systems (SDSs) and microprocessor driver systems (MDSs). SDSs use photosensors in order to detect a change in light sources and convert it into an electrical signal, which is used to obtain the position of the sun. However, there are tracking errors when the sensors cannot produce an electrical signal due to low solar radiation levels produced by the presence of passing clouds or contamination in the air. MDSs use microprocessors and computer systems to execute mathematical equations based on solar position algorithms and the current date and time to determine the exact position of the sun. MDSs are cheaper than SDSs; however, there is no feedback to verify the position of the sun, and tracking errors may appear due to the precision of the solar position algorithm. Several algorithms with different levels of complexity and accuracy can be found in the literature [3], where the use of a more precise solar position algorithm increases both the accuracy and the computational effort of the system.

Central tower power plants use two-axis sun reflectors called heliostats, which reflect the solar irradiance into a collector tower. Every heliostat has a local control which drives two motors connected to reduction gears, where the trend is to give greater autonomy to the central control by increasing

the intelligence of the local control of each heliostat. Additionally, in solar plants, there are changing dynamics due to non-linearities and uncertainties that traditional PID controllers cannot handle. This is because a PID controller may produce high oscillations when it is tightly tuned, and the dynamics of the process varies due to changes in the operating conditions. Hence, the use of more efficient control strategies results in better responses [1]. An FLC is a good alternative to traditional PID controllers, because it can deal with non-linear systems and can be designed by using the knowledge of a human operator without knowing the mathematical model of the system. Although the FLC does not have a better response in time domain than a PID controller, this later one cannot be applied to systems which have a quick change of parameters because it would require to adjust the value of its control gains [4].

FLCs [5–10] and hybrid PID-FLCs [4,11–16] have been applied to control the position of DC motors, showing a good output response and better performance against traditional PID controllers. Furthermore, FLCs have been implemented to control two-axis sun trackers for photovoltaic systems. Yousef [17] was the first to develop a PC-based FLC algorithm to control a two-axis photo-voltaic (PV) solar panel. Afterwards, the FLC has been implemented in different platforms and devices such as microcontrollers [18–20], DSPs [21], personal computers (PCs) [22–25] and field-programmable gate arrays (FPGAs) [26].

Finally, the FLC has also been applied in orientation control of heliostats. Ardehali and Emam [27] performed a comparison between a classical PI and PID controller, a PI-FLC and a PID-FLC for the orientation control of a laboratory-scale heliostat with two mirrors of 0.9 m × 0.7 m, two 15 W DC motors, and a data acquisition system with 20 ms sampling time. The FLC uses three membership functions in order to adjust the PID controller gains. The results showed that PI-FLC presented reductions in the overshoot and better performance than the other controllers. Zeghoudi and Chermitti [28] and Zeghoudi et al. [29] used the Matlab environment in order to simulate the orientation of a heliostat by using an FLC with two different rule bases, comparing the output response with a PI, a PID, a PI-FLC and a neural controller. The results showed a better output response for the FLC compared with the other controllers. Additionally, the FLC with fewer rules showed a better output response to step changes than the FLC with the bigger rule base. Bedaouche et al. [30] simulated the position control of two DC motors in order to modify the orientation of a heliostat by using a PID controller self-adjusted by an FLC. The FLC adjusts each PID controller gain by using an individual rule base of forty-nine rules and the error and change of error values. The results showed a faster output response and a smaller overshoot than a classic PID controller. Jirasuwankul and Manop [31] applied an FLC to control the orientation of a lab-scale heliostat with two stepper motors by using a micro-step driver. The position of the heliostat is obtained by using image processing of the reflected solar radiation on the target. The results showed a good performance of the FLC; however, there are tracking errors when the system cannot process the image due to passing clouds.

Nevertheless, the works cited above have only been presented in simulations and small-scale models. Considering the aforementioned, the objective of this paper is to describe the design and implementation of a two-axis STS for the orientation control of a real-scale azimuth-elevation heliostat by using an FLC implemented on a low-cost microcontroller-based embedded system. The comparison between the FLC and a PID controller has also been done.

## 2. Heliostat Orientation Control

The orientation control system is presented in Figure 1. The control system modifies the angular position of two DC motors connected to the axes of the heliostat through two worm drive mechanisms to guide the heliostat to the desired position. A microcontroller unit (MCU) calculates the position of the sun and the desired angles of the heliostat in order to reflect the solar radiation on a specific target by using the geographic position of the heliostat and the current time and date values. Afterwards, a position control algorithm calculates the error between the desired and current angular position of the heliostat axes by using two rotary encoders in order to obtain a control signal which orients the heliostat by using two motor drivers that allow the bidirectional control of the DC motors.

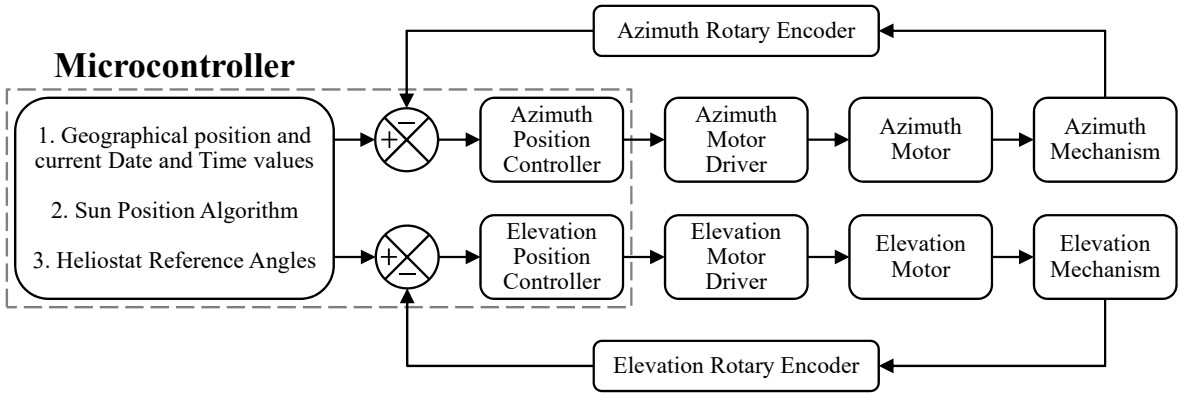

**Figure 1.** Orientation control diagram.

*2.1. DC Motor Mathematical Model*

A DC motor can be described by using the equivalent model shown in Figure 2. The reduced transfer function of the armature-controlled DC motor is given by (1) [32].

$$G(s) = \frac{\Theta(s)}{V_a(s)} = \frac{k_m}{s[R_a(Js + b) + k_b k_m]} \tag{1}$$

where $k_m$ represents the motor torque constant and $k_b$ represents the back electromotive-force constant.

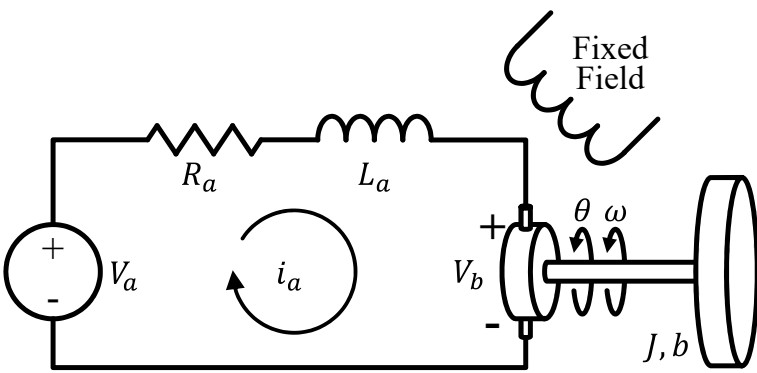

**Figure 2.** Equivalent model of a permanent magnet brushed DC motor.

The mathematical model of the DC motor can be estimated by using the step signal response method with the motor speed response under a fixed voltage. The transfer function of the position and speed model can be described by (2) and (3).

$$\frac{\Theta(s)}{V_a(s)} = \frac{\frac{k_m}{JR_a}}{s\left(s + \frac{bR_a + k_b k_m}{JR_a}\right)} = \frac{C_k}{s(s + C_p)} \tag{2}$$

$$\frac{\Omega(s)}{V_a(s)} = \frac{C_k}{s + C_p} = \frac{\frac{C_k}{C_p}}{\frac{1}{C_p}s + 1} = \frac{K}{\tau s + 1} \tag{3}$$

where $C_k$ and $C_p$ are fixed parameters, $\tau$ represents the time constant, and $K$ represents the steady-state gain of the system.

The steady-state gain is the ratio of the output and the input in steady-state [33] and is given by (4).

$$K = \frac{\omega_s}{u_{step}} = \frac{C_k}{C_p} \tag{4}$$

where $\omega_s$ represents the steady speed of the DC motor and $u_{step}$ represents the step input signal.

Finally, the transfer function of the DC motor is given by (5).

$$G(s) = \frac{\Theta(s)}{V_a(s)} = \frac{\frac{\omega_s}{u_{step}}}{s\left(s + \frac{1}{\tau}\right)} \tag{5}$$

### 2.2. Control Algorithms

#### 2.2.1. PID Controller

The PID controller is the most commonly used in industrial applications due to its simple structure. However, its linear nature makes it not very suitable for non-linear systems. It is a control technique which reduces the error ($e(t)$) of a system using three control gains (Proportional, Integral and Derivative) in a mathematical operation to produce a control output ($u(t)$). The equation for the PID controller in the time domain is described by (6) [32].

$$u(t) = K_p e(t) + K_i \int e(t)dt + K_d \frac{de(t)}{dt} \tag{6}$$

When the controller is digital, it can be approximated with a backward difference and a sum for the derivative and integral terms [4], respectively. The digital PID controller is given by (7).

$$u(n) = K_p e(n) + K_i \sum_{j=1}^{n} e(j)T_s + K_d \frac{e(n) - e(n-1)}{T_s} \tag{7}$$

where $n$ and $T_s$ represent the number of the sample and the sample time of the digital system.

#### 2.2.2. Fuzzy Logic Controller

An FLC uses the experience of an expert instead of the mathematical model of the system to control a plant, and it can deal with complex non-linear systems with unknown mathematical models. The controller produces a control signal using four blocks [34]: fuzzification, inference engine, rule base and defuzzification, as shown in Figure 3.

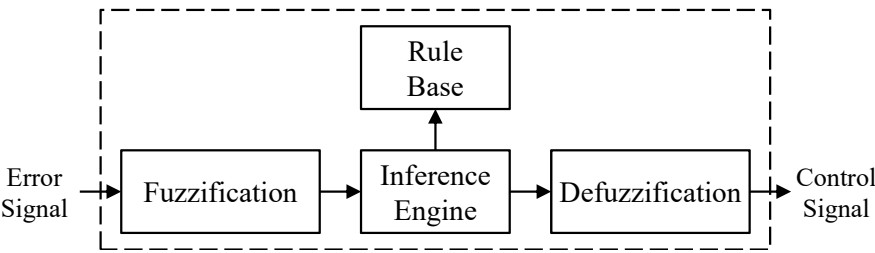

**Figure 3.** Components of the FLC.

The FLC is graphically shown in Figure 4. The fuzzification module converts the input values into fuzzy sets using the singleton fuzzification, which evaluates the membership value of the input value. The inference mechanism determines the values of the output fuzzy sets by using an "if–then" rule base, which describes the relationship between the input and output variables based on their linguistic terms. In Mamdani fuzzy systems, the rule base determines the output fuzzy set value taking the minimum value of the combination of two or more input fuzzy set values as a consequence of one rule in the rule base. Finally, the defuzzification module gets a scalar value by combining the scaled output fuzzy sets values.

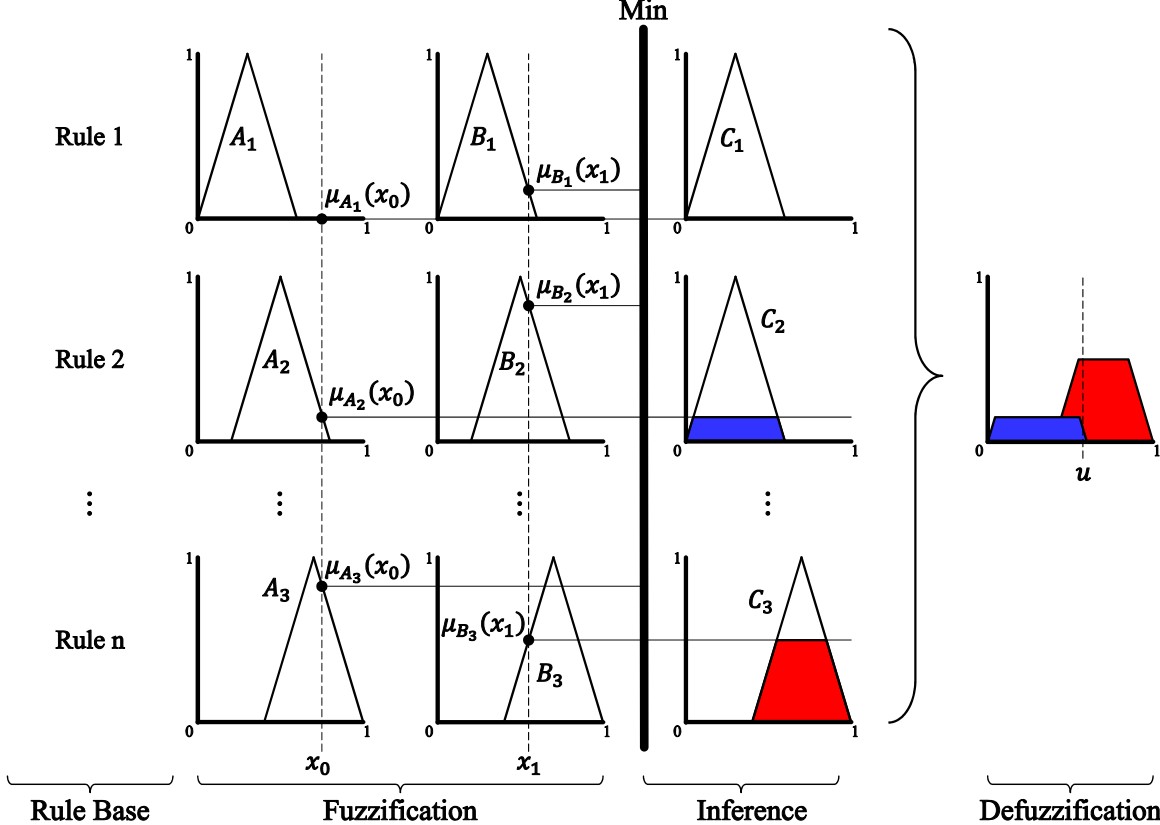

**Figure 4.** Fuzzy logic controller algorithm.

The reduction of the number of fuzzy rules as long as they express a similar relationship decreases the computational effort and memory requirements used in the implementation of the controller [35]. Therefore, it is necessary to eliminate the less critical rules in order to obtain faster controller actions [36].

The defuzzification module is another component that can be modified in order to obtain a fast response of the controller. There are many defuzzification methods proposed in the literature [34]. The center of gravity method (CoG), also called the center of area method (CoA), is the most widely used of all the defuzzification methods. Nonetheless, this method has a very high computational effort. The CoG method calculates the area under the combined output fuzzy sets by sampling them between the minimum and maximum values of the output fuzzy sets, as shown in Figure 5a. The drawback of the CoG is that it requires more samples to obtain a more accurate output value. The output value of the CoG defuzzification method is given by (8).

$$u = \frac{\sum\limits_{i=1}^{n} \mu(x_i) x_i}{\sum\limits_{i=1}^{n} x_i} \tag{8}$$

where $x_i$ represents a value between the minimum and maximum values of the scaled output fuzzy sets, and $n$ represents the number of the samples.

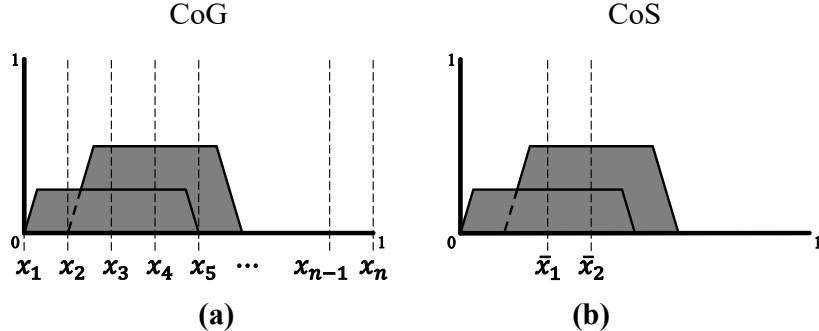

**Figure 5.** Center of Gravity (**a**) and Center of Sums (**b**) defuzzification methods.

Another defuzzification method is the center of sums method (CoS), which is a fast method because of its computational simplicity [37]. This method calculates the average between the centroid and the area of each scaled output fuzzy set. The drawback of the CoS is that the intersecting areas are added twice, as shown in Figure 5b. The output value of the CoS defuzzification method is given by (9).

$$u = \frac{\sum\limits_{i=1}^{n} \mu(\overline{x}_i) A_i}{\sum\limits_{i=1}^{n} A_i} \tag{9}$$

where $\overline{x}_i$ and $A_i$ represent the centroid and the scaled area of the output fuzzy set $i$, and $n$ represents the number of the output fuzzy sets.

### 2.3. Sun Position and Heliostat Angles

Due to the fact that the relative position of the sun in the sky changes throughout the day, it is necessary to use a solar tracker in order to know the location of the sun at any time. The position of the sun with respect to the observer can be described by a reference system of horizontal coordinates using two angles: the azimuth angle and the elevation angle [1]. The angles of the solar vector $\overrightarrow{S}$ are denoted by $A_s$ and $E_s$, respectively, as shown in Figure 6a. The azimuth angle is measured in relation to the South (0°), and it is negative to the East (−90°) and positive to the West (90°). The elevation angle of the sun ranges from the horizon (0°) to the zenith (90°).

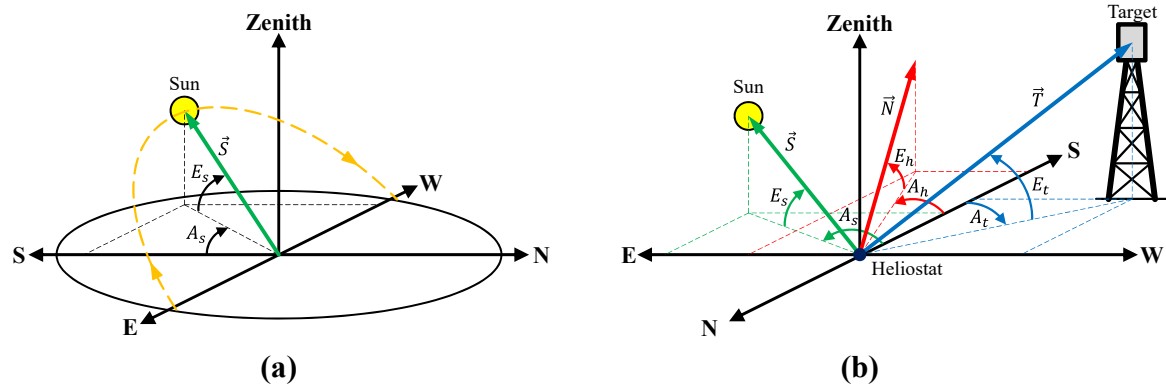

**Figure 6.** Solar vector (**a**) and vectors and angles of the heliostat (**b**).

For the heliostat to reflect the solar irradiance towards the central receiver, the heliostat surface normal vector $\overrightarrow{N}$ must be the bisector of the angle formed by the fixed vector pointing to the receiver from the reflective surface of the heliostat $\overrightarrow{T}$ and the solar vector [1] (Figure 6b). The azimuth and elevation angles of the solar vector are given by the Grena [38] algorithm, which has a maximum error

of 0.0027°. The solar position algorithm takes the geographical coordinates of the heliostat and the current date and time of the day as input data. The algorithm also uses the monthly average local values of temperature and atmospheric pressure to calculate the atmospheric refraction correction of the elevation angle of the sun. The azimuth and elevation angles of the target vector are obtained by using spherical coordinates. The normal vector is obtained by the addition of the unit vectors of the solar and target vectors.

$$\vec{N} = \left( \begin{array}{ccc} \hat{S}_x + \hat{T}_x & \hat{S}_y + \hat{T}_y & \hat{S}_z + \hat{T}_z \end{array} \right) \tag{10}$$

where $\hat{S}$ and $\hat{T}$ are given by (11) and (12).

$$\hat{S} = \left( \begin{array}{ccc} sin(A_s)cos(E_s), & cos(A_s)cos(E_s), & sin(E_s) \end{array} \right) \tag{11}$$

$$\hat{T} = \left( \begin{array}{ccc} sin(A_t)cos(E_t), & cos(A_t)cos(E_t), & sin(E_t) \end{array} \right) \tag{12}$$

Finally, the azimuth and the elevation angles of the normal vector are given by (13) and (14).

$$A_h = tan^{-1} \left( \frac{\vec{N}_y}{\vec{N}_x} \right) \tag{13}$$

$$E_h = tan^{-1} \left( \frac{\vec{N}_z}{\sqrt{\vec{N}_x^2 + \vec{N}_y^2}} \right) \tag{14}$$

*2.4. Embedded System*

The block diagram of the embedded system is shown in Figure 7. The heliostat orientation control is implemented in a dsPIC33EP256MU806 MCU running with a clock frequency of 48 MHz. The current date and time values are given by a real-time clock (RTC) model DS1307 with I2C (Inter-Integrated Circuit) serial interface protocol. Two H-Bridge motor drivers built with four bipolar junction transistors (tip135 and tip136) are connected to the embedded system in order to change the direction of rotation of the DC motors by using an external power supply and two control signals from the MCU for each DC motor, whereas the feedback signal of the controller is given by two single-turn absolute rotary encoders model CAS60RS12A10SGG with synchronous serial interface (SSI) protocol and 12 bits of resolution (4096 pulses per revolution). Both rotary encoders are connected to the axes of the heliostat in order to obtain the real position of the heliostat. The system also contains an alphanumeric LCD Display to visualize the initial controller parameters, an analog thumb joystick for the manual heliostat control, a UART block to send data to a computer to perform graphical analysis, and a programming port ICSP (In-Circuit Serial Programming).

The algorithm of the embedded system was designed and developed by using CCS C Compiler software and is shown in Figure 8. All fixed values are read from a database at the start of the program. These values include the geographical position of the heliostat, distance to the target, local weather record, configuration data of the microcontroller peripherals, and parameters and grogram functions of the control algorithms. Afterwards, the program runs an infinite loop and waits for a start command to move the heliostat to the desired position. The program uses four 16-bit timers to generate software interrupts at fixed intervals of time in order to operate different components of the system. Timer1 generates a 200 kHz frequency square signal in order to communicate the MCU with the rotary encoders through the SSI protocol, timer2 establishes the period of the PWM signal which controls the speed and position of the DC motors by using the motor drivers, timer3 performs a software interrupt every 10 ms for the sampling time of the control algorithms, and timer4 performs a

software interrupt every second in order to send the data to the UART block and read the time and date values from the RTC.

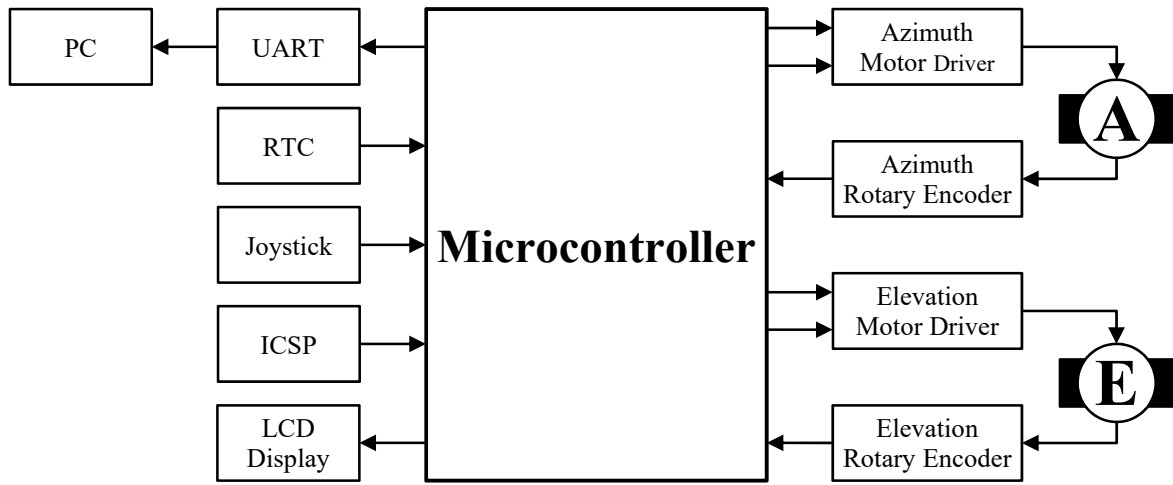

**Figure 7.** Block diagram of the embedded system.

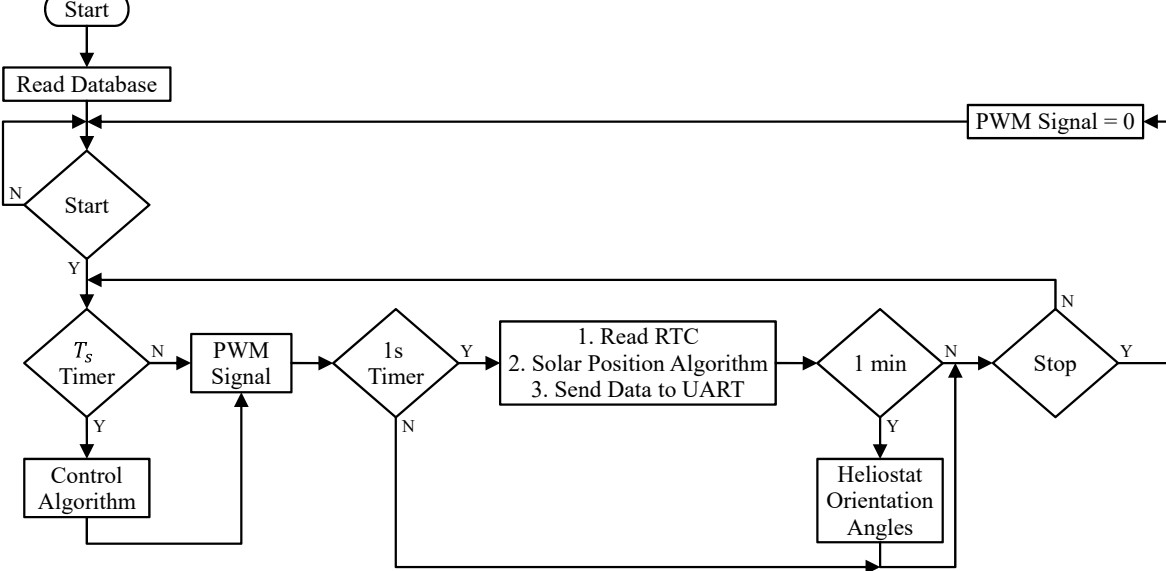

**Figure 8.** Embedded system algorithm.

Once the start command is received, the program reads the date and time values from the RTC to compute the position of the sun by using the solar position algorithm. Afterwards, the program calculates the desired position of the heliostat to determine the reference values of the control algorithm. Finally, the program calculates the value of the error between the reference values and the position of the heliostat axes which is given by the rotary encoders and determines the control signal of each DC motor by using a program function that takes the error value and returns the values of the voltage that must be supplied to each DC motor. The voltage values are converted into duty cycle values of the PWM signals, which are supplied to the motor drivers in order to move the heliostat to the desired position by adjusting the angular position of each DC motor.

The position of the sun is calculated every second when the value of the RTU changes. However, the reference values of the control algorithm can be set in a fixed period without producing a significant error in the incidence of solar irradiance in the target. Therefore, the desired position of the heliostat is calculated every minute.

The control algorithms are shown in Figures 9–11, for the PID controller and FLC with the CoG and Cos defuzzification methods, respectively.

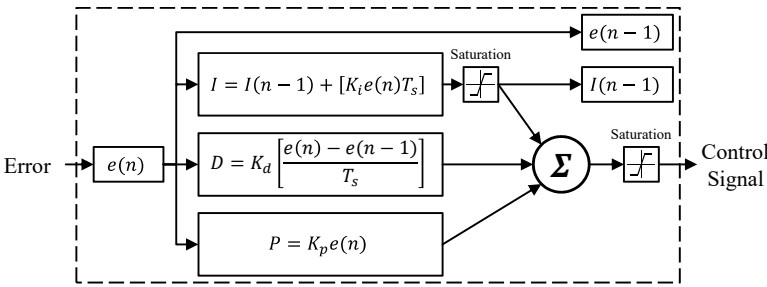

**Figure 9.** PID controller algorithm.

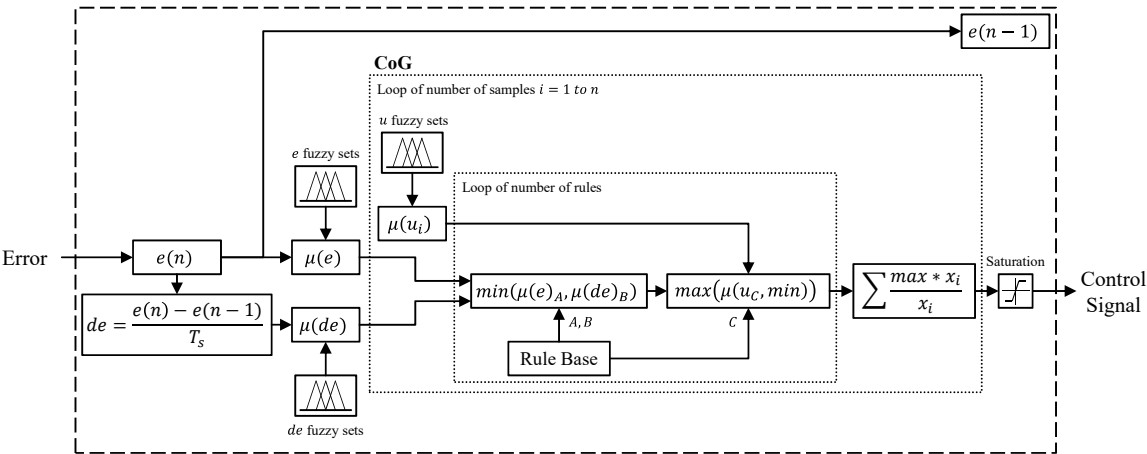

**Figure 10.** Fuzzy logic controller algorithm with the center of gravity defuzzification method.

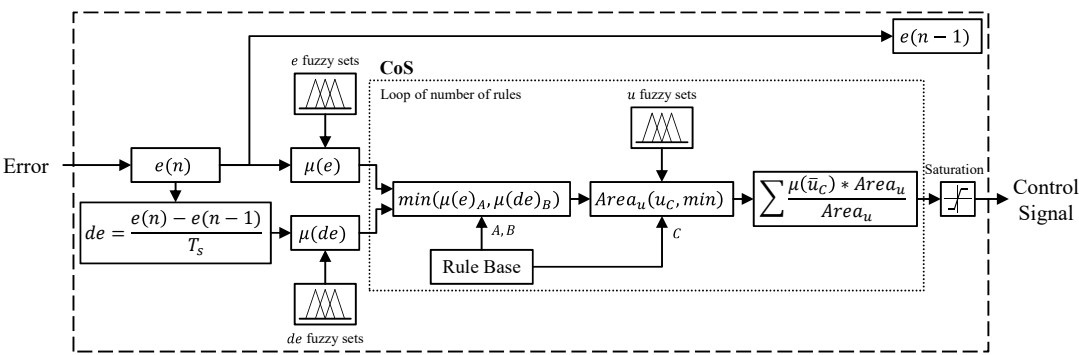

**Figure 11.** Fuzzy logic controller algorithm with the center of sums defuzzification method.

The PID controller uses the error value to obtain the control signal by using the control gains loaded from the database and Equation (7). A saturation block is used on the integral term to limit its value and obtain a faster response at changes in the error value.

The FLC algorithm obtains the value of the change of error by using the error value and a backward difference, in order to evaluate the input fuzzy sets. Afterwards, the rule base determines the output fuzzy set that corresponds to the values of the error and changes of error and combines it according to the defuzzification method in order to obtain the control signal. The CoG defuzzification method executes a loop for the number of samples that evaluates the output fuzzy sets. In each iteration of the loop, all the rules are evaluated by using another loop for the number of rules in order to obtain the maximum value of the evaluated output fuzzy sets in the sample value, as shown in Figure 5a. Finally,

all resulting values are added to obtain the output value by using Equation (8). The CoS defuzzification method only executes one loop, calculating the output value by using values of the scaled area and the centroid of each output fuzzy set, as indicated in Equation (9). The values of the centroid of the output fuzzy sets are calculated once at the beginning of the program and do not change.

The values of the error and integral term are saved in order to calculate the terms used in the next sample of the control algorithms. There is also a saturation block to limit the output signal of the control algorithms at the rated voltage value of the DC motors.

### 2.4.1. Controller Parameters

The algorithms of the FLC and PID controller were designed for the position control of the DC motors at no load. Both controllers were tuned to accomplish with the design parameters of 10 ms sampling time and 100 ms of rising time without overshoot for the smallest change in the reference signal in order to reduce the energy consumed by the DC motors when the heliostat is moving [27].

Figure 12 shows the block diagram of the FLC. It is a two-input and one-output controller, three fuzzy sets in each input and output signal, a rule base with nine "if–then" rules, a Mamdani inference engine and the CoS defuzzification method. Additionally, there are two processing blocks due to the difference between the fuzzy sets values and the values of the input and output signals. The processing values are given by (15)–(17).

$$e* = \frac{e}{\pi} \tag{15}$$

$$de* = \frac{de T_s}{\pi} \tag{16}$$

$$u* = u V_{max} \tag{17}$$

where $e*$, $de*$ and $u*$ represent the processing values of the input and output signals, and $V_{max}$ represents the maximum voltage signal of the DC motors.

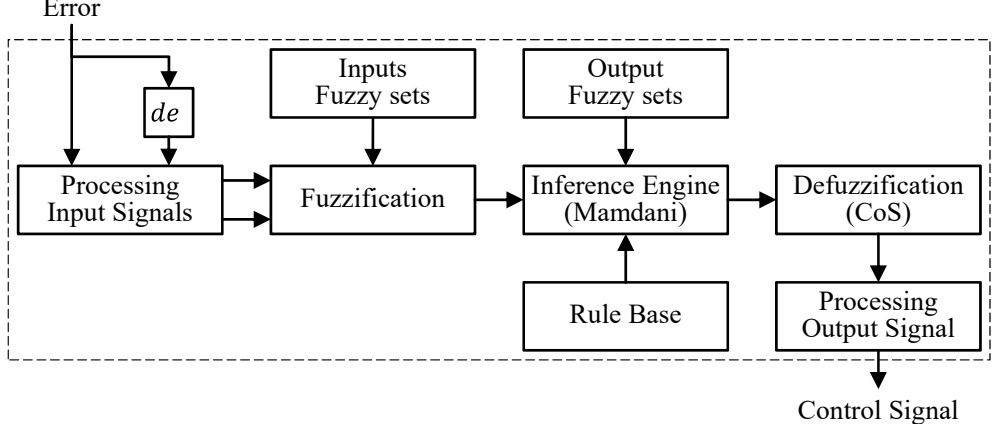

**Figure 12.** Block diagram of the fuzzy logic controller.

The values of the fuzzy sets and the rule base of the FLC are shown in Figure 13 and Table 1, where the negative, middle, and positive values are denoted by the linguistic variables $N$, $Z$ and $P$, respectively. The number and values of the fuzzy sets were selected in order to the control signal of the FLC can modify the position of the DC motor due to the smallest change of the error with a low computational effort. The symmetric shape of the fuzzy sets allows the controller to modify the direction of rotation of the DC motors with the same amplitude of the control signal, which corresponds to the values of the error and change of error.

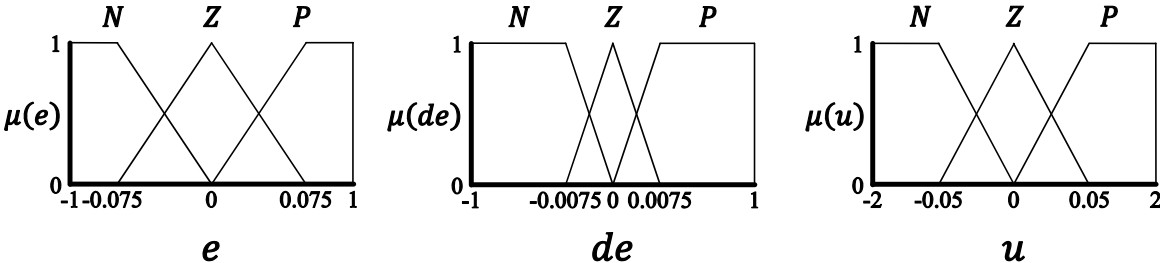

**Figure 13.** Fuzzy sets of the fuzzy logic controller.

**Table 1.** Rule base of the fuzzy logic controller.

| de \ e | N | Z | P |
|---|---|---|---|
| N | N | N | Z |
| Z | N | Z | P |
| P | Z | P | P |

The resulted control surface of the FLC is presented in Figure 14, showing the relationship between the inputs and the output as a consequence of the values of the fuzzy sets, the if–then rule base, and the CoS defuzzification method. The output value of the defuzzification varies from −1 to 1; therefore, using Equation (17), the DC motor supply voltage ranges from $-V_{max}$ to $V_{max}$.

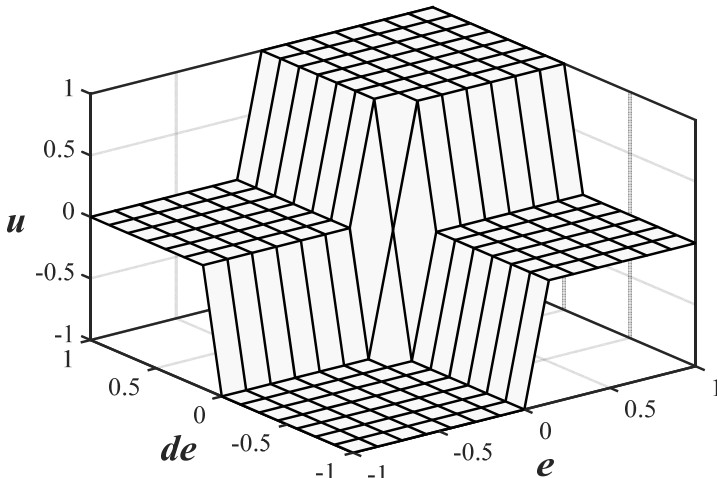

**Figure 14.** FLC control surface.

For the PID controller, the transfer function of the DC motor is estimated by using the step signal response method in order to obtain the control gains of the PID controller to comply with the design parameters. The angular velocity can be approximated by using a discrete derivative term, as shown in Equation (7). The angular velocity is given by (18).

$$\omega(n) = \frac{\theta(n) - \theta(n-1)}{T_s} \tag{18}$$

The step response of the DC motor and the transfer function parameters are shown in Figure 15, where $\omega = 0.5369 \; \frac{rad}{s} = 5.126$ rpm is approximately the rated speed reported in the DC motor datasheet, as shown in Table 3.

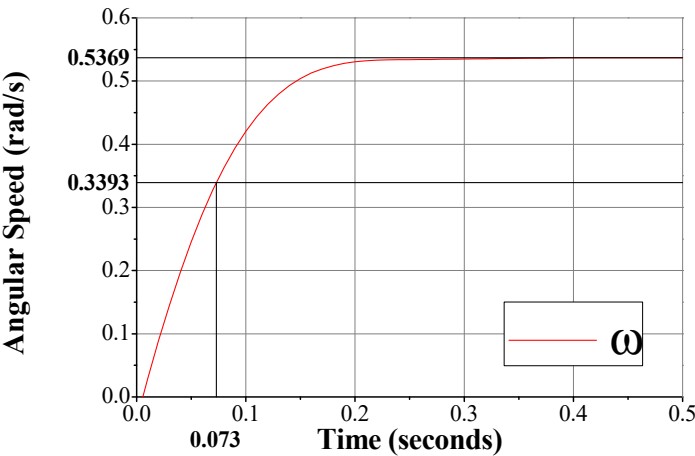

**Figure 15.** Step response of the DC motor.

The obtained mathematical model of the DC motor at no load is described by (19).

$$G_M(s) = \frac{\Theta_M(s)}{V_{aM}(s)} = \frac{0.31275}{s^2 + 13.68925s} \tag{19}$$

Finally, the control gains of the PID controller were obtained by using the Matlab Sisotool Toolbox. A rising time of 0.075 s and an overshoot of less than 5% were chosen as conditions of the output signal of the PID controller in order to accomplish with the design requirements and produce a smooth control signal to reduce the energy consumption for the DC motors. The control gains of the PID controller are shown in Table 2.

**Table 2.** Control gains of the PID controller for the DC motor at no load.

| $K_p$ | $K_i$ | $K_d$ |
|-------|-------|-------|
| 2250.0 | 0.025 | 110.0 |

The transfer functions of the DC motors connected to the heliostat axes were also obtained with the same method. The mathematical models of the azimuth and elevation axes are described by (20) and (21), respectively.

$$G_A(s) = \frac{\Theta_A(s)}{V_{aA}(s)} = \frac{0.01316}{s^2 + 4.03225s} \tag{20}$$

$$G_E(s) = \frac{\Theta_E(s)}{V_{aE}(s)} = \frac{0.02417}{s^2 + 7.40740s} \tag{21}$$

2.4.2. Setpoint Values

Because of the position of the sun in the sky changes by 1 degree every 4 min, it is not necessary to modify the orientation of the heliostat every second of the day. Therefore, the values of the reference angles are discretized every minute, as shown in Figure 16 for the parameters of Table 4.

To reduce the error due to the resolution of the rotary encoders, the discrete reference value is converted from radians to encoder steps and is rounded to the closest integer value to obtain a final reference value that corresponds to a value in the encoder steps. Therefore, when the heliostat angles reach the desired position, the error signal will be zero. The values of the error between the final and desired reference of the heliostat axes are shown in Figure 17.

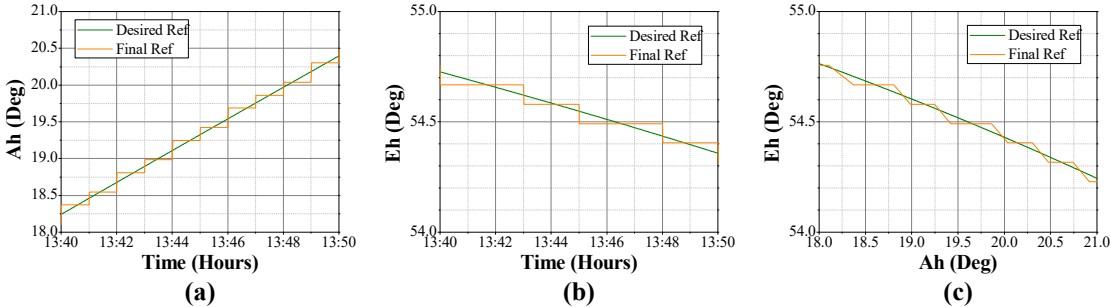

**Figure 16.** Setpoint values of the orientation control for the azimuth axis (**a**), the elevation axis (**b**) and both axes (**c**).

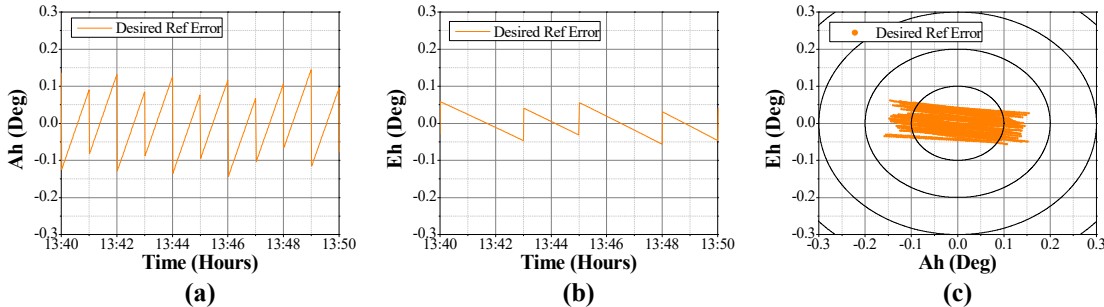

**Figure 17.** Final reference error values of the orientation control for the azimuth axis (**a**), the elevation axis (**b**) and both axes (**c**).

## 3. Results and Discussion

The orientation control system was implemented in the heliostat shown in Figure 18, whereas the printed circuit board (PCB) of the embedded system and the motor driver are shown in Figure 19. It is an azimuth–elevation mechanism heliostat, with a worm drive mechanism driven by a DC gear motor model ZYT6590-01 at each axis. The heliostat has a gap which allows directing the facets to the ground. The parameters of the heliostat and the DC motors are presented in Table 3.

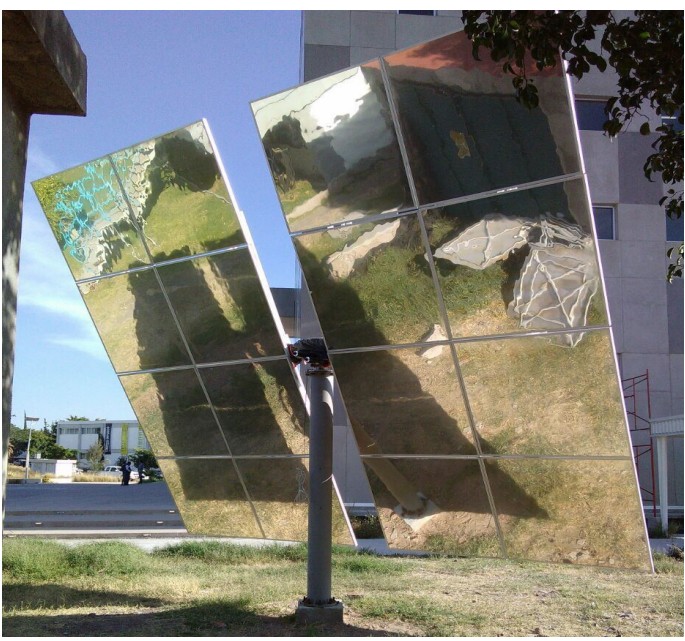

**Figure 18.** Heliostat.

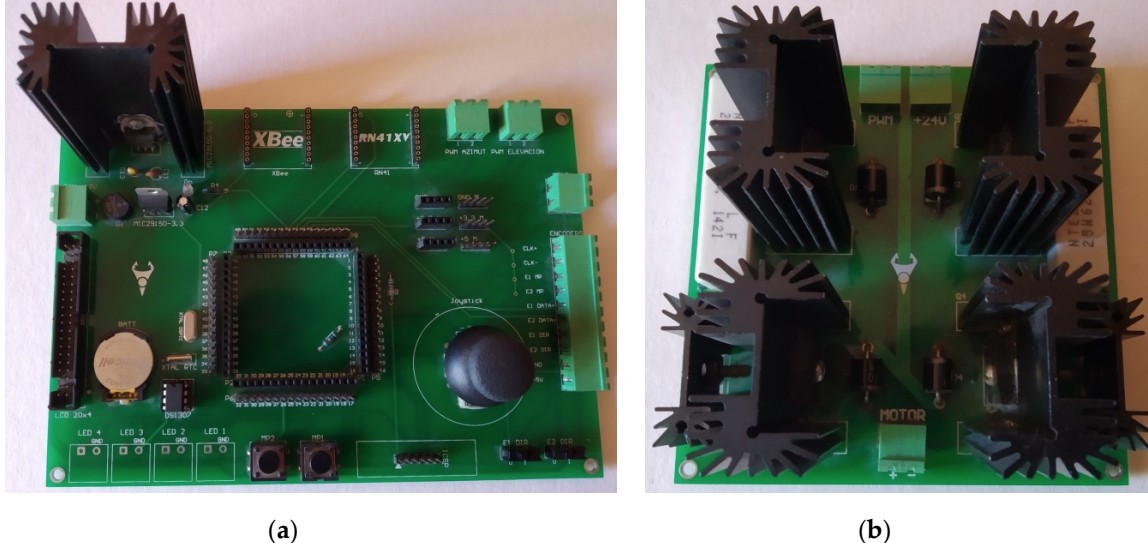

(**a**)                    (**b**)

**Figure 19.** Printed circuit board of the embedded system (**a**) and the motor driver (**b**).

**Table 3.** Parameters of the heliostat and DC motors.

| Parameter | Value | Unit |
|---|---|---|
| Total height | 5.24 | m |
| Pedestal height | 2.85 | m |
| Elevation axis length | 4.43 | m |
| Gap between support frames | 0.70 | m |
| Number of facets | 16 | - |
| Mirror face size | $1.2 \times 1.2$ | m |
| Heliostat mirror area | 23 | $m^2$ |
| DC Motors Rated Voltage | 24 | V |
| DC Motors Rated Current | ≤5 | A |
| DC Motors Rated Torque | 100 | N·m |
| DC Motors No Load Speed | 5 | rpm |
| DC Motors Gear Ratio | 710.5 | - |

As mentioned already, the control algorithms were designed for the position control of a DC motor at no load. Afterwards, the control algorithms were implemented in the orientation control of the heliostat using the same controller parameters of the position control of the DC motor at no load.

Figure 20 shows the comparison of the consumption time of the PID controller (Figure 20a) and the FLC using the CoS (Figure 20b) and the CoG (Figure 20c) defuzzification methods, where the period of the signals represents the sampling time of the control algorithms. The results show that the FLC with the CoG defuzzification method does not accomplish with the design parameters because of its computational complexity.

The output response of the control algorithms for the position control of a DC motor at no load is shown in Figure 21 for the minimum change in the reference value of 0.087 degrees (1.533 mrad) and a reference value of 180 degrees ($\pi$ rad). Both control algorithms accomplish with the design parameters for the position control of the DC motor at no load. However, Figure 21d shows that the FLC control signal decreases when the position of the DC motor is reaching the reference value.

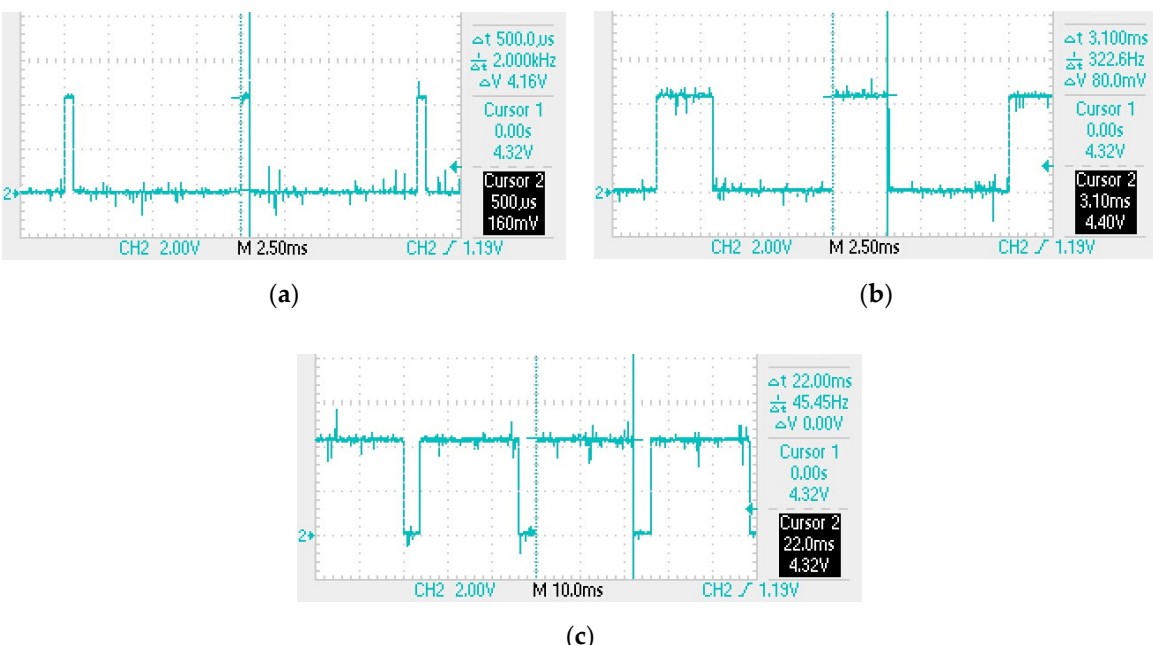

**Figure 20.** Consumption time of the PID controller (**a**), and the fuzzy logic controller with the CoS (**b**) and the CoG (**c**) defuzzification methods.

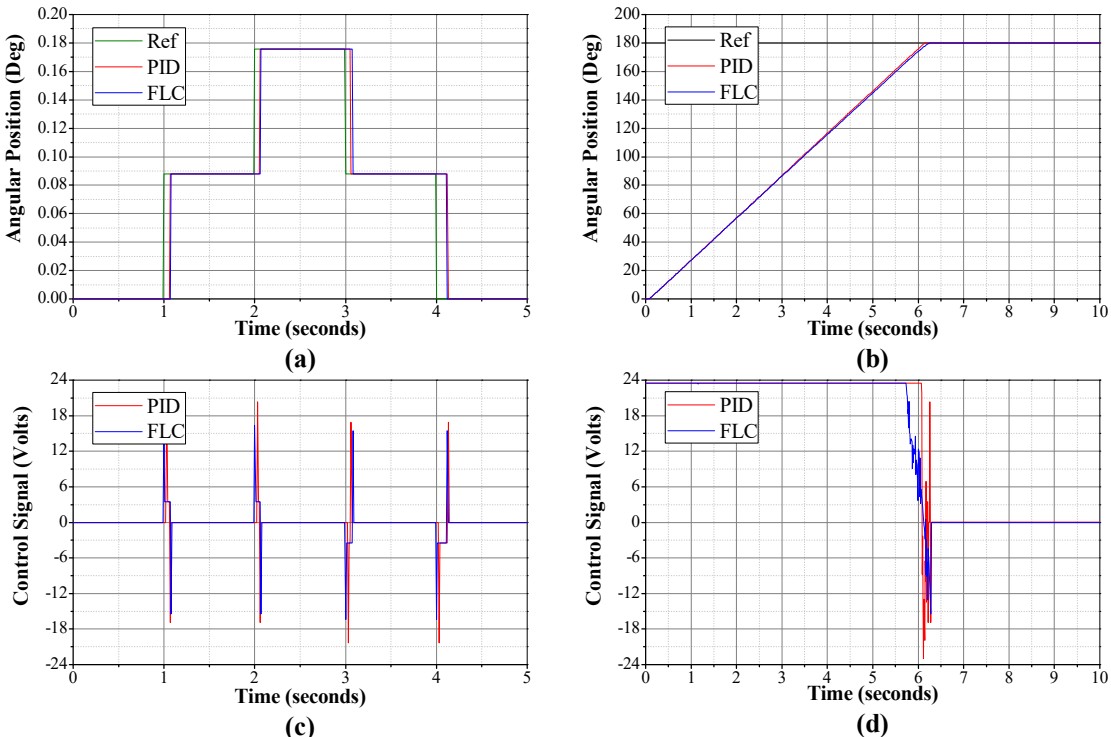

**Figure 21.** Output response (**a**) and control signal (**c**) of the DC motor at no load at a minimum reference value. Output response (**b**) and control signal (**d**) of the DC motor at no load at a reference value of 180 degrees ($\pi$ rad).

Finally, Figures 22 and 23 show the output response of the control algorithms for the orientation control for the DC motors at no load and the axes of the heliostat, respectively. The desired angles of the heliostat were calculated using the parameters of Table 4, whereas the error values are shown in Table 5.

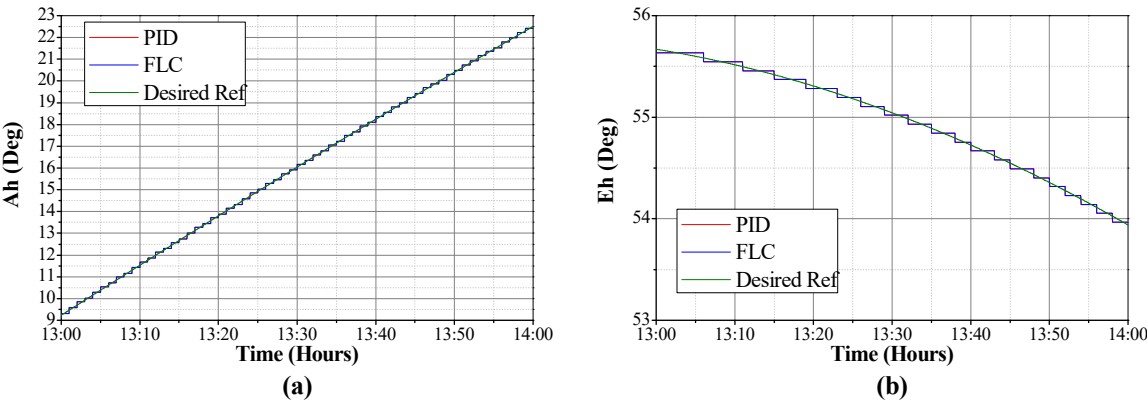

**Figure 22.** Output response of the orientation control of the DC motors at no load for the azimuth axis (**a**) and the elevation axis (**b**).

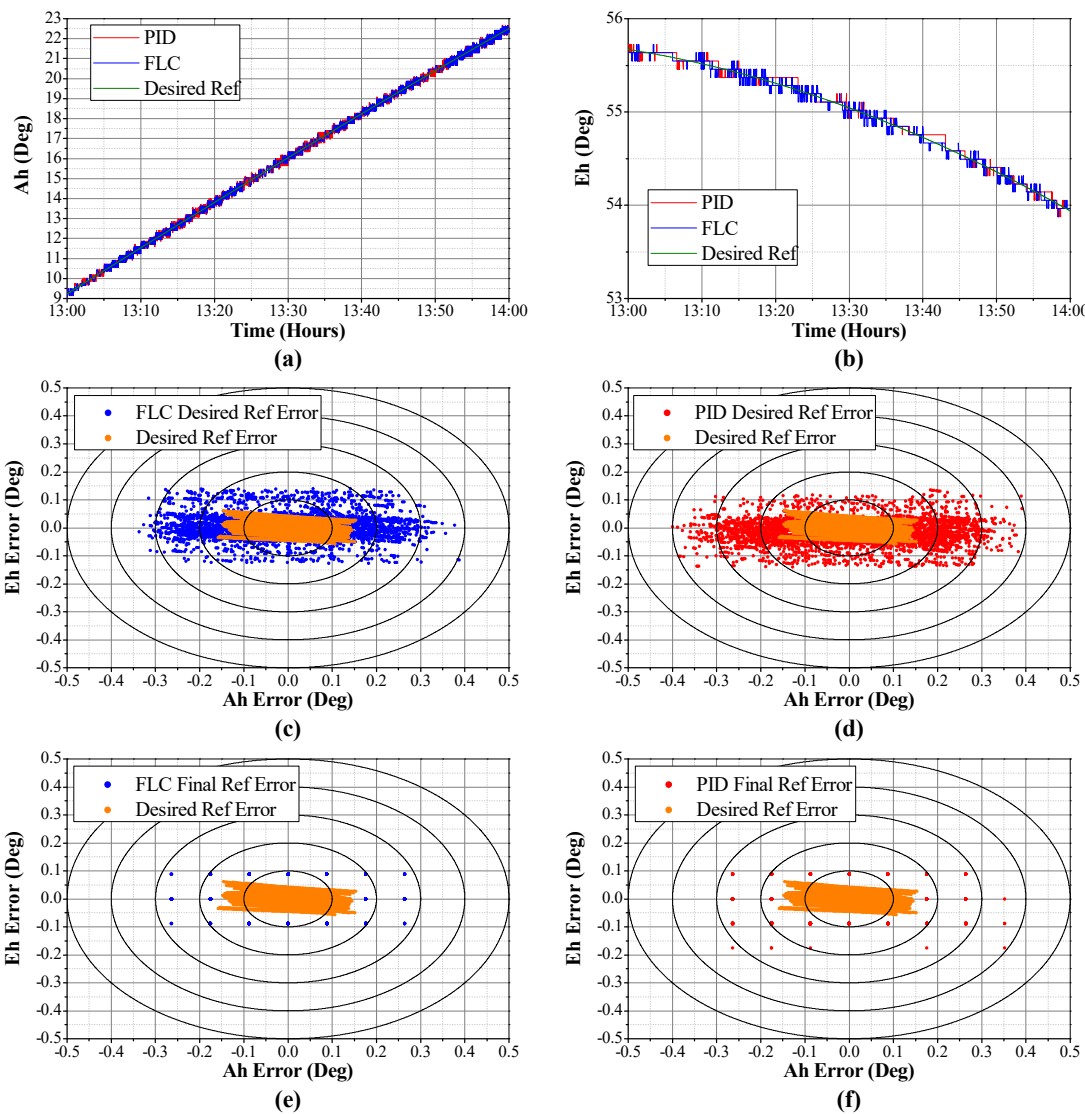

**Figure 23.** Output response of the orientation control for the azimuth axis (**a**) and the elevation axis (**b**) of the heliostat. Desired reference error values of the orientation control of the heliostat for the fuzzy logic controller (**c**) and the PID controller (**d**). Final reference error values of the orientation control of the heliostat for the fuzzy logic controller (**e**) and the PID controller (**f**).

**Table 4.** Parameters of the orientation control test.

| Parameter | Value |
|---|---|
| Date | Friday, 13 September 2019 |
| Time | 13:00:00–14:00:00 |
| Latitude | 20.590636° N |
| Longitude | 100.413226° W |
| Monthly Mean Atmospheric Pressure | 819.795 mbar |
| Monthly Mean Temperature | 20.3 °C |
| Maximum Wind Speed | 8 m/s (28.8 km/h) |
| Target Height | 30.0 m |
| Heliostat Height | 2.85 m |
| East-West distance to the target | 15 m East |
| North-South distance to the target | 35 m North |

**Table 5.** Reference error values of the orientation control.

| Parameter | | Final Ref MSE | Desired Ref MSE |
|---|---|---|---|
| DC Motor at no load | PID Azimuth | 0.0° | 0.068610° |
| | PID Elevation | 0.0° | 0.026349° |
| | FLC Azimuth | 0.0° | 0.068610° |
| | FLC Elevation | 0.0° | 0.026349° |
| Heliostat | PID Azimuth | 0.153941° | 0.168669° |
| | PID Elevation | 0.051032° | 0.048347° |
| | FLC Azimuth | 0.131647° | 0.146435° |
| | FLC Elevation | 0.039328° | 0.047251° |

The experimental results show a similar Mean Squared Error (MSE) for the orientation control of the DC motors at no load and a similar output response between the orientation control of the heliostat and the final reference value for the FLC (Figure 23a) and the PID controller (Figure 23b), despite the load of the wind over the mechanical structure and the backlash in the axis mechanisms. However, for the orientation control of the heliostat, the FLC shows less dispersed error values (Figure 23c) and smaller final reference error values (Figure 23e) than the PID controller (Figure 23d,f).

## 4. Conclusions

The orientation control of a heliostat using an FLC was implemented on an embedded system based on a low-cost microcontroller. Also, the comparison against a traditional PID controller was performed. The advantage of the FLC is the fact that it is not necessary to know the mathematical model of the system, because it only uses the experience of an operator, which is easy to incorporate into the controller.

The results show that both controllers exhibit a similar output response for the position control of a DC motor. However, the FLC has a better performance than the PID controller for the orientation control of the heliostat by using the same control parameters for the position control of the DC motor at no load. The FLC has higher flexibility since it is robust in front of changes in the dynamics of the process, whereas for a better output response of the PID controller, the control gains must be tuned for the mathematical models of the heliostat axes. The results also exhibit a smaller MSE of the FLC compared to the PID controller for the orientation control of the heliostat by using only a nine-rule rule base and a fuzzy set of three membership functions in each input and output signal in order to reduce the computational effort of the controller. Additionally, the center of the sums defuzzification method complies with the design parameter of 10 ms sample time, showing a faster response than the center of the gravity defuzzification method.

In a central tower power plant that uses traditional PID controllers for the orientation control of the heliostats, the control gains of the controller of all the heliostats must be adjusted in order to

avoid oscillations due to wrong controller parameters tuning. Therefore, the proposed control system can be applied in order to control an entire heliostat field by using the same controller parameters for all the heliostat. The system can also be adjusted to control other sun tracking systems, such as a photovoltaic, solar dish, or parabolic trough systems, which only need the solar tracker system.

**Author Contributions:** Conceptualization, E.S.-P. and M.T.-A.; Methodology, E.S.-P. and R.V.C.-S.; Project administration, M.T.-A.; Software, E.S.-P.; Validation, E.S.-P. and R.V.C.-S. All authors have read and agreed to the published version of the manuscript.

**Funding:** This research received no external funding.

**Acknowledgments:** The authors would like to thank Consejo Nacional de Ciencia y Tecnología (CONACYT-México) for supporting this research.

**Conflicts of Interest:** The authors declare no conflict of interest.

## Abbreviations

| | |
|---|---|
| CoG | Center of Gravity |
| CoS | Center of Sums |
| DSP | Digital Signal Processor |
| FLC | Fuzzy Logic Controller |
| FPGA | Field-Programmable Gate Array |
| LCD | Liquid Crystal Display |
| MCU | Microcontroller Unit |
| MDS | Microprocessor Driver System |
| MSE | Mean Squared Error |
| PC | Personal Computer |
| PCB | Printed Circuit Board |
| PID | Proportional–Integral–Derivative |
| PV | Photo-Voltaic |
| PWM | Pulse Width Modulation |
| RTC | Real-Time Clock |
| SDS | Sensor Driver System |
| STS | Sun Tracking System |
| UART | Universal Asynchronous Receiver-Transmitter |
| $\theta$ | Angular position of the DC motor |
| $\omega$ | Angular velocity of the DC motor |
| $\tau$ | Time constant of the system |
| $K$ | Steady-state gain of the system |
| $e$ | Controller error signal |
| $de$ | Controller change of error signal |
| $u$ | Controller output signal |
| $T_s$ | Controller sampling time |
| $V_{max}$ | Controller maximum output voltage |
| $\vec{S}$ | Solar vector |
| $\vec{T}$ | Target vector |
| $\vec{N}$ | Normal vector of the heliostat |
| $\hat{S}$ | Solar unit vector |
| $\hat{T}$ | Target unit vector |
| $A_s$ | Solar vector azimuth angle |
| $E_s$ | Solar vector elevation angle |
| $A_t$ | Target vector azimuth angle |
| $E_t$ | Target vector elevation angle |
| $A_h$ | Heliostat azimuth angle |
| $E_h$ | Heliostat elevation angle |

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
