# Peer review of "Development of a DSP Microcontroller-Based Fuzzy Logic Controller for Heliostat Orientation Control"

_applsci, doi:10.3390/app10051598_

Round 1

Reviewer 1 Report

The authors present a Fuzzy logic based Heliostat orientation controller using microcontroller. The paper is well presented but some more clarification is required for the following:

Details of microcontroller implementation are missing. Some state-diagrams, flow charts can help..  A part from no requirement of model, FLC does not prove any better to PID. In case where model is also available what benefits can FLC bring in the proposed system. Authors need to properly justify it.

The aforementioned issues may please be addressed before considering this manuscript any further.

Reviewer 2 Report

In equation (2) you need to replace "j" to "J"

Reviewer 3 Report

Development of a DSP Microcontroller-based fuzzy logic controller for heliostat orientation control

The authors present a fuzzy control application to move two heliostat orientation motors (angular position). They provide a DSP based method implementing it on a dsPiC series device from Microchip(R).

To evaluate the results, they have compared the conclusion regarding a traditional PID controller equally microcontroller-based.

Comments to the author:

The manuscript is well structured, clear and uses appropriate language. It is very easy to read.

However, I would like to address some major points:

1- It could consider that the solution proposed is very widespread. This reviewer's biggest concern is identifying what is truly novel about this work.  It gives an excellent both motor device and Fuzzy controller fundamentals review and formulation.

Nevertheless, the authors do not declare explicit what is the novelty in this reported work regarding previous contributions from other researchers, including the same authors. They must locate their technical contribution.

2- Sections 2.1 and 2.2 present information usually provided in numerous text-book. If the authors re-consider it, they could be references uniquely for the sake of brevity.

3- Figure N.1 shows a four-variable control system: two angular estimated sun position and two angular references (encoders feedback). Possibly, different strategies for a rectangular plate control should be applied (facets  spatial distribution). That leads to different PID adjustments (a tuning for each motor) or/and no-similar Fuzzy sets. This issue was not considered in this article.

4- Section 2.4 presents the embedded module. However, the text does not contribute anything about whether it is the author's custom design or it is a commercial platform-based.

The authors must remember the cited Applied Sciences Journal aims, as, “….The full experimental details must be provided so that the results can be reproduced. “

The authors should report about the design supplying schematics, or pictures, system features, references, etc. None information is reported concerning to the electronic driver for the included motor (ZTT6590, 5 to 24 V and 5 A) or the gear-box angular resolution (where is fixed the encoder shaft?). The paper does not report the binary resolution of the A/D and D/A conversions, as well as the numerical resolution, which is also unknown.

In general, the article does not report anything about software implementations. It does not specify the language utilized to program the DSP device (assembler, C, Phyton, etc.).  Moreover, it does not specify the technique followed for the software implementation of the PID or the fuzzy Controller.

Particularly, regarding the fuzzy implementation case, the paper does not report about the software method implementation (use of known specific Fuzzy libraries or own custom algorithms). How is algorithmically evaluate the CoG or CoS?  Can the PID be manually tuned for different checking?

5- Section 2.4.1: Both the design of the FLC controller and the PID does not consider the mechanical load effect (considered in Section 3, lines 289-294). Obviously, in an environment situated control system, the effects of "noise" on the controlled elements (referring table 3 and Fig. 17 structure, wind on the mirrors array, the asymmetric effect of gravity on the motor structure depending on the axis and facets location, gear gaps, etc.) may be considered, to technically enrich the article. Table 3 could include other structure parameters as inertia values for each axis, and the viscous friction parameters expected.

In this same section, fuzzy sets features are introduced without any argument. It is not explained why the membership function distribution of each set is divided in this way (three similar fuzzy sets for both inputs and similar membership areas/shapes, applied without justification). Likewise, the choice of only three sets to represent the output angular variation could be poor. Lines 128-130 do not justify the code simplification. Today devices as dsPIC have sufficient memory and resources to carry out these implementation kinds, using ASM or C languages.

6- Section 3: The test was carried out using no-load motors, and then, the system was tested using the heliostat as a mechanical load. Regarding the no-load case, the authors must elaborate on why azimuth and elevation axis (Fig. 16 and Fig.20) present different error values respect the time (for the same motor type, and using the same controller parameters).

Referring now to Figure 21 (different errors for azimuth and elevation): it could be possible this error difference due to the structure is not necessarily symmetric concerning each axis. It is correct?

7- Conclusions: Previous work has already concluded the advantage of the fuzzy controller both non-linear systems as well as systems which it is difficult to analyze the control equations.

The PID tuning was performed at the "no-load" experiment. For this reason, at the "load" mode tuning, the PID will give a lower settling error concerning the result shown in Fig21d and Fig21f.

Therefore the authors could highlight the advantage of fuzzy control for the specific case of the solar power tower. For a solar plant, there would be a large number of reflective elements and PIDs that would expressly require a specific tuning process, depending on every spatial position. For situations like this, a fuzzy control will present advantages according to the results previously obtained in the work.

Reviewer 4 Report

Suggestions:

The literature review seems incomplete. Please provide a more comprehensive review of different tracking methods, such as the review provided in this article:  https://ieeexplore.ieee.org/document/8614997 Provided a comparative analysis for MDS vs. SDS. What are the advantages and disadvantages of each method? Time constant is a well-known concept in physics and engineering. I do not see the need of explaining this concept and having a Figure wasted for explaining it.  Similarly, I feel that too much has been said on the basics of DC motors, PID control, and Fuzzy Logic. These are all well-established concepts in electrical engineering.  Use of English in this paper is proper but minor errors exist. Proofreading is recommended. Time consumption and mathematical complexity of the two implemented methods (FLC and PID) should be compared. Maybe parts of the microcontroller code or its pseudocode can be presented as an appendix. What is the basis for tuning the PID controller for rising time of 0.075 s and an overshoot of less than 5%? There is so much similarity between this work and previous work of the authors: https://www.mdpi.com/2076-3417/9/15/2966/htm What is the novelty of this work compared to the following publications? https://ieeexplore.ieee.org/document/8015479 , https://ieeexplore.ieee.org/document/8477396 , https://www.tandfonline.com/doi/pdf/10.1007/s12543-011-0081-x 

Round 2

Reviewer 3 Report

The authors have made a great effort in improving and readjusting the text according to the indications received.

Reviewer 4 Report

The authors have applied my previous suggestions.

  • Minor formatting and grammar errors exist. Please proofread before publication.
  • I still feel the literature review could be further improved using the references I provided in my previous review.